# OBSERVER UNCERTAINTY OF LEARNING IN GAMES FROM A COVARIANCE PERSPECTIVE

## ABSTRACT

We investigate the accuracy of prediction in deterministic learning dynamics of zero-sum games with random initializations, specifically focusing on observer uncertainty and its relationship to the evolution of covariances. Zero-sum games are a prominent field of interest in machine learning due to their various applications, such as Generative Adversarial Networks. Concurrently, the accuracy of observation in dynamical systems from mechanics has long been a classic subject of investigation since the discovery of the Heisenberg Uncertainty Principle. This principle employs covariance and standard deviation of particle states to measure observation accuracy. In this study, we bring these two approaches together to analyze the follow-the-regularized-leader (FTRL) algorithm in two-player zero-sum games. We provide growth rates of covariance information for continuous-time FTRL, as well as its two canonical discretization methods (Euler and symplectic). Our analysis and experiments shows that employing symplectic discretization enhances the accuracy of prediction in learning dynamics.

## 1 INTRODUCTION

In recent years understanding the behavior of learning algorithms in games has attracted increasing interests from the machine learning community (Lanctot et al., 2017; Yang & Wang, 2020). Follow-the-Regularized-Leader (FTRL) algorithm (Abernethy et al., 2009; Shalev-Shwartz et al., 2012),[1] arguably the most well known class of no-regret dynamics, plays a prominent role in analysis of behavior of learning algorithms. The behavior of such online learning algorithm in zero-sum games has been a particularly intense object of study as zero-sum games related to numerous recent applications and advances in AI such as, achieving super-human performance in Go (Silver et al., 2016), Poker (Brown & Sandholm, 2018) and Generative Adversarial Networks (GANs) (Goodfellow et al., 2014) to name a few.

It is well known that in any zero-sum games, the time-average of no-regret dynamics converges to Nash equilibria. However, if restricted to time-average convergence, one might confront obstacles in understanding the day-to-day behavior, which is of significance in many of the aforementioned applications. Unfortunately, FTRL dynamics (both in their standard discrete-time implementation as well as their continuous-time approximations) do not to converge in zero-sum games with their behavior being complex and even formally chaotic(Mertikopoulos et al., 2018; Piliouras & Shamma, 2014; Cheung & Piliouras, 2019; 2020). This makes predicting their behavior and outcome of the learning algorithms difficult or even impossible as an infinitesimal small perturbation of the initial conditions may be very quickly amplified by the learning dynamics. This raises the following two challenges for us:

*How can we qualitatively study the evolution of uncertainty in game dynamics? Could subtle differences in implementation of well known online learning dynamics result to significant differential in their performance from an uncertainty perspective?*

The most relevant paper in this direction is (Cheung et al., 2022), which recently introduced a framework to study the evolution of uncertainty of Multiplicative Weights Updates (MWU) (i.e. FTRL with negative entropy regularizer) in zero-sum games and variants thereof. The key idea is not to focus on the evolution of a single initial condition (e.g., whether it converges to equilibrium), but instead study the behavior of a probability distribution/measure over a set of initial conditions. This

---

[1]FTRL is also known as Dual Averaging (Nesterov, 2009).

"input uncertainty" is well motivated both from a game-theoretic perspective (observer uncertainty about the agents' initial beliefs) as well as from a Machine Learning perspective (system initialization by sampling from a data distribution). Given this formulation they used differential entropy, the analogue of standard Shannon entropy for continuous variables (Michalowicz et al., 2013), as their metric of uncertainty and showed that differential entropy grows linearly fast in two players zero-sum games quantifying the amount of excess information an outsider observed must gather to keep track of the uncertain system evolution.

In this work, on top of differential entropy, we will be introducing a novel perspective in quantifying uncertainty, motivated by the classic mechanics formulation of FTRL (Bailey & Piliouras, 2019; Balduzzi et al., 2018) and the uncertainty principle from quantum mechanics. Specifically, given a random system initialization (in the relevant system state-space) we will be focusing on the evolution of variances of the related random variables. This perspective is closely related to the well known as *Heisenberg Uncertainty Principle* in quantum mechanics, which captures that one cannot measure the momentum and position of a microscopic particle accurately at the same time, i.e., $\Delta p \Delta q \geq h$. The Hamiltonian formulation of FTRL endows the cumulative strategy and payoff of each agent the roles of position $q$ and momentum $p$ of each particle. Once the initialization is randomized, the deterministic learning dynamics still makes the cumulative strategy and payoff random variables whose mean and covariance matrix is related to that of the initialization. Therefore, it is natural to ask how the variances of cumulative strategies and payoffs evolve with FTRL dynamics. As what is implied from Heisenberg Uncertainty, information on standard deviations (variances, covariance matrices) measures the *accuracy of prediction* in learning dynamics.

**Our contributions.** In this paper we study differential entropy and covariances in FTRL for two-player zero-sum games. We investigate the evolution of observer uncertainty measured by these two quantities in two popular discretization schemes, Euler and symplectic discretization. We establish the following results as main contributions.

- Differential entropy remains constant when two players take turns to update their mixed strategies (Alternating play), see Proposition 4.2;

- We propose covariance matrix as an uncertainty measurement which captures both simultaneous and alternating plays, with rate of increasing calculated concretely in Euclidean regularized FTRL, see Theorem 5.1;

- For FTRL with general regularizers, a Heisenberg type inequality on variances of cumulative strategy and payoff is obtained, i.e., $\Delta X_{i,\alpha} \Delta y_{i,\alpha} \geq$ positive constant. This inequality indicates a general uncertainty principle, tradeoff between accuracy in strategy spaces versus payoff spaces for game dynamics, see Therorem 5.2.

**The technological innovations.** The techniques used in drawing the above conclusions come from different areas. The theoretical framework for analyzing the uncertainty evolution is the classic mechanical formulation of games (Bailey & Piliouras, 2019; Wibisono et al., 2022). To demonstrate that differential entropy is constant in alternating plays, we utilize the volume preservation property of Symplectic discretization. Furthermore, the intuition in deriving the covariance evolution of Symplectic discretization is from (Wang, 1994), and the proof combine tools from matrix analysis such as the Jordan normal form. The uncertainty inequality for general FTRL is a consequence of a classic result from symplectic geometry known as non-squeezing theorem and variance analysis methods from multivariate statistics.

**Motivation and the place of this paper in the literature.** The current paper is situated within the research on the dynamics of no-regret online learning algorithms and represents a natural extension of previous works that address this problem from the perspectives of volume or uncertainty evolution, such as (Cheung & Piliouras, 2019; 2020; Cheung et al., 2022). A main motivation for this work is to address certain drawbacks in these works and further expand upon their results. Volume analysis was introduced as a tool to study the evolution of a region of initial conditions under the simultaneous MWU algorithm in (Cheung & Piliouras, 2019; 2020). From this perspective, it was discovered that the simultaneous MWU algorithm exhibits what is known as Lyapunov chaos behaviors. Similar tools are also used to study day-to-day behaviors of no-regret learning algorithms in (Flokas et al., 2020).

Driven by the need of considering the initial uncertainty of learning in game, (Cheung et al., 2022) purposed to use differential entropy as a measurement of uncertainty, and extended volume analysis to calculate how differential entropy evolve under simultaneous MWU algorithm. On this stage, it is nature to ask : *Is differential entropy a good measure of uncertainty in games dynamics? More especially, as alternating playing is more common in real word games, can the results in (Cheung et al., 2022) be extended from simultaneous playing to alternating playing ?* Motivated by above questions, we have found that differential entropy is not a suitable measure for capturing changes in uncertainty during alternating play. Therefore, the next question is: *Which concept is more appropriate for measuring uncertainty evolution in game dynamics?* The idea of using variance as a measure of uncertainty originates from the Heisenberg uncertainty principle, where covariance and standard deviation are employed to quantify the uncertainty in the quantum dynamics. Moreover, since we care about day-to-day behavior and the risk in decision making, variance and covariance has prominent advantage in quantifying risk (it has been well studied in statistics and finance, data science), this is why we study uncertainty with variance and covariance.

**Further related works.** The family of Follow-the-Regularized-Leader algorithms in the continuous time and several their properties such as volume-preserving and existing Poincaré recurrence are investigated in (Mertikopoulos et al., 2018). However, it turns out the behaviors of discrete time algorithms is quite different from their continuous time counterpart. In the case of simultaneous gradient descent ascent with constant step size, the strategy orbits will not converge and keep away form its stationary points with an exponential rate (Mescheder et al., 2018; Gidel et al., 2019) and in the case of simultaneous multiplicative weights updates, the volumes of strategy orbits have fast growth rate and converge to the boundary of constrains (Bailey & Piliouras, 2018; Cheung, 2018; Cheung & Piliouras, 2020). Alternating gradient-descent-ascent and its several variants are studied in (Gidel et al., 2019; Bailey et al., 2020; Zhang & Yu, 2020; Yang et al., 2020) and (Wibisono et al., 2022) consider more general alternating mirror descent algorithms in constrained case.There are several other forms of uncertainties in game theory and min-max optimization. Games with random payoff matrixes and their applications in mathematical biology are studied in (Gao, 2013; Duong et al., 2019b;a). Another kind of uncertainty comes from the stochastic algorithms (Chavdarova et al., 2019; Luo et al., 2020; Yang et al., 2020), where reducing the variance introduced by the algorithms is a main focus. Our results are orthogonal to aforementioned ones.

## 2 PRELIMINARIES

**Learning in games.** A two agent zero-sum game consists of two agent $\mathcal{N} = \{1, 2\}$, where agent $i$ selects a strategy from the strategy space (or primal space) $\mathcal{X}_i \subset \mathbb{R}^{n_i}$ and $n_i$ represents the number of actions available to agent $i$. Typically, $\mathcal{X}_i$ is chosen to be $\mathbb{R}^{n_i}$, which we called the unconstrained zero sum game, or it is chosen to be the simplex $\Delta_i = \{x \mid \sum_{s=1}^{n_i} x_s = 1, x_s \geq 0\}$. Utilities of both agents are determined via payoff matrix $A^{(ij)} \in \mathbb{R}^{n_i \times n_j}$, and in a zero-sum game, the pay off matrix satisfy $A^{(ij)} = -A^{(ji)}$. For convenience, we will also use $A$ to refer to $A^{(12)}$, and thus $A^{(21)} = -A^\top$. Given that agent $i$ selects strategy $x_i \in \mathcal{X}_i \subset \mathbb{R}^{n_i}$, agent 1 receives utility $u_1(x_1, x_2) = \langle x_1, Ax_2 \rangle$, and agent 2 receives utility $u_2(x_2, x_1) = -\langle x_2, A^\top x_1 \rangle$. Naturally agents want to maximize their utility resulting the following max-min problem:

$$\max_{x_1 \in \mathcal{X}_1} \min_{x_2 \in \mathcal{X}_2} x_1^\top A x_2. \qquad \text{(Zero-Sum Game)}$$

**Follow-the-Regularized-Leader.** Follow-the-Regularized-Leader algorithm (FTRL) is a widely used class of no-regret online learning algorithms. In continuous time FTRL, at time $t$, agent $i$ updates strategies $x_i(t)$ based on the cumulative payoff vector $y_i(t)$,

$$y_i(t) = y_i(0) + \int_0^t A^{(ij)} x_j(s) ds \qquad \text{(Continuous FTRL)}$$

$$x_i(t) = \arg\max_{x_i \in \mathcal{X}_i} \{\langle x_i, y_i(t) \rangle - h_i(x_i)\}$$

where $h_i$ is a strongly convex function, which is called the regularizer. It is also well known that $x_i(t) = \nabla h_i^*(y_i(t))$, where $h_i^*(y_i) = \max_{x_i \in \mathcal{X}_i} \{\langle x_i, y_i \rangle - h_i(x_i)\}$ is the convex conjugate of $h_i$ (Shalev-Shwartz & Singer, 2006). Gradient descent ascent (GDA) and multiplicative weights updates (MWU) are two of the most well known special cases of FTRL algorithms. For unconstrained GDA, the regularizers are chosen to be the Euclidean norm, i.e., $h_i(x_i) = \|x_i\|^2$ and $\mathcal{X}_i = \mathbb{R}^{n_i}$. For

MWU, the regularizers are chosen to be negative entropy, i.e., $h_i(x_i) = \sum_{j \in [n_i]} x_{i,j} \ln(x_{i,j})$ and $\mathcal{X}_i = \{x_{i,j} \mid \sum_{j=1}^{n_i} x_{i,j} = 1, x_{i,j} > 0\}$. In the following, we will use GDA to refer to unconstrained GDA and omit the word unconstrained.

In discrete time, FTRL has two kinds of implementations : simultaneous and alternating. In the case of GDA and MWU, the update rules with step size $\eta$ are :

$$x_1^t = x_1^{t-1} + \eta A x_2^{t-1}, \quad x_2^t = x_2^{t-1} - \eta A^\top x_1^{t-1}, \tag{GDA}$$

$$x_2^t = x_2^{t-1} - \eta A^\top x_1^{t-1}, \quad x_1^t = x_1^{t-1} + \eta A x_2^t, \tag{AltGDA}$$

and

$$x_1^t = \left( \frac{x_{1,s}^{t-1} e^{\eta(A x_2^{t-1})_s}}{\sum_{j=1}^{n_1} x_{1,j}^{t-1} e^{\eta(A x_2^{t-1})_j}} \right)_{s=1}^{n_1}, \quad x_2^t = \left( \frac{x_{2,s}^{t-1} e^{\eta(-A^\top x_1^{t-1})_s}}{\sum_{j=1}^{n_2} x_{2,j}^{t-1} e^{\eta(-A^\top x_1^{t-1})_j}} \right)_{s=1}^{n_2}, \tag{MWU}$$

$$x_2^t = \left( \frac{x_{2,s}^{t-1} e^{\eta(-A^\top x_1^{t-1})_s}}{\sum_{j=1}^{n_2} x_{2,j}^{t-1} e^{\eta(-A^\top x_1^{t-1})_j}} \right)_{s=1}^{n_2}, \quad x_1^t = \left( \frac{x_{1,s}^{t-1} e^{\eta(A x_2^t)_s}}{\sum_{j=1}^{n_1} x_{1,j}^{t-1} e^{\eta(A x_2^t)_j}} \right)_{s=1}^{n_1}. \tag{AltMWU}$$

**Dynamical system.** A system of ordinary differential equations $\dot{x} = f(x)$ where $f : \mathbb{R}^n \to \mathbb{R}^n$ is a differentiable dynamical system. $f(x)$ is called the vector field of the dynamical system. If $f$ is Lipschitz continuous, there exists a continuous map $\varphi(t, x_0) : \mathbb{R} \times \mathbb{R}^n \to \mathbb{R}^n$ such that for all $x_0 \in \mathbb{R}^n$, $\varphi(t, x_0)$ is the unique solution of the initial condition problem $\{\dot{x} = f(x), x(0) = x_0\}$. The solution $\varphi(t, x_0)$ is called a ***trajectory*** or ***orbit*** of the dynamical system.

**Hamiltonian systems.** A Hamiltonian system is a class of differential equations describing the evolution of momentums and positions of particles by a scalar function $H(X, Y, t)$ ($H(X, Y)$ for time-independent case) called Hamiltonian function. The state of the system, the momentum $Y = (y_1, ..., y_n)^\top$ and position $X = (x_1, ..., x_n)^\top$ evolves according to the following Hamilton's equations:

$$\frac{dx_i}{dt} = \frac{\partial H}{\partial y_i}, \quad \frac{dy_i}{dt} = -\frac{\partial H}{\partial x_i}, \quad \text{for} \quad i \in [n]. \tag{1}$$

The solution $\varphi(t, \cdot)$ of a Hamiltonian system is called a ***symplectic*** map which is a special case of volume-preserving maps, thus the absolute value of determinant of the Jacobian matrix equals to 1.

**Linear ordinary differential equation.** Let $P$ be an $n \times n$ matrix. Then for a given $z_0 \in \mathbb{R}^n$, the initial value problem

$$\frac{dz}{dt} = Pz, \quad z(0) = z_0 \tag{2}$$

has a unique solution given by $z(t) = e^{Pt} z_0$, where the exponential operator $e^{Pt}$ is defined as $e^{Pt} \triangleq \sum_{k=0}^{\infty} \frac{P^k t^k}{k!}$.

**Remark 2.1.** *In general, one cannot expect for a shortcut to understand linear systems only from the formal expression $z(t) = e^{Pt} z_0$. Matrix exponential function is fundamentally different from usual exponential function is several aspects. For example, in general $e^{P_1} e^{P_2} \neq e^{P_2} e^{P_1}$ and this makes direct calculation of $e^{Pt}$ impossible. In practice even computing matrix exponential numerically for large matrices is a challenging task ([Moler & Van Loan, 2003](#)).*

### 2.1 MEASURE OF OBSERVER UNCERTAINTY

**Differential Entropy.** The concept of differential entropy was introduced by Shannon ([Shannon, 1948](#)) as a measure of the uncertainty associated with a continuous probability distribution. It has now found applications in many fields ([Neeser & Massey, 1993](#); [Garbaczewski, 2006](#)).

For a random vector $X \in \mathbb{R}^n$ with probability density function $g(x)$ supported on $\mathcal{X} \subset \mathbb{R}^n$, the differential entropy of $X$ is defined as

$$S(X) = -\int_{\mathcal{X}} g(x) \log g(x) dx. \tag{Differential Entropy}$$

**Covariances of random vectors.** Given a random vector $X = (x_1, ..., x_m)^\top$ such that every $x_i$ is a random variable with finite variance and expected value, the covariance matrix $P(X) \in \mathbb{R}^{m \times m}$ of $X$ is a symmetric and positive semi-definite square matrix whose $(i, j)$ entry is the covariance, i.e.,

$$\text{Cov}(x_i, x_j) = \mathbb{E}[(x_i - \mathbb{E}(x_i))(x_j - \mathbb{E}(x_j))].$$

Note that the diagonal elements of $P(x)$ are variances of $\{x_1, ..., x_m\}$. Moreover, for a matrix $M \in \mathbb{R}^{m \times m}$, we have $P(MX) = MP(X)M^\top$.

In general, the differential entropy of a random variable provides a lower bound on the determinant of its covariance matrix. Precisely, if a random vector $X \in \mathbb{R}^m$ has zero mean and covariance matrix $P(X)$, then $S(X) \leq \frac{1}{2} \log\left((2\pi e)^n \det P(X)\right)$, and equality holds if and only if $X$ has a joint Gaussian distribution.

## 3 SETUP

In this section we leverage the power of the classic mechanic formulation of game dynamics (Balduzzi et al., 2018; Bailey & Piliouras, 2019). To study the strategy-payoff evolution from the perspective of each agent, it is convenient to apply the Hamiltonian formulation of the continuous time FTRL. In this subsection, we will establish the equivalence between discretization of the Hamiltonian system induced by continuous time FTRL and direct discretization of FTRL, where the latter leads to the Gradient Descent-Ascent (with Euclidean regularizer) or Multiplicative Weights Update (with entropy regularizer) in the strategy space $\mathcal{X}_i$. We will make this correspondence clear in the sequel.

**Euler discretization.** Given an ordinary differential equation $\dot{x} = f(x)$ with initial condition $x(t_0)$ at time $t_0$, the Euler discretization begin the process by setting $x_0 = x(t_0)$, next choose a step size $\eta$ and set $t_n = t_0 + n\eta$, then the Euler discretization $\phi_\eta(\cdot)$ is defined by $x_{n+1} = \phi_\eta(x_n) = x_n + \eta f(x_n)$. The value $x_{n+1}$ is an approximation of the solution of $\dot{x} = f(x)$ at time $t_{n+1}$.

**Symplectic discretization.** Given a Hamiltonian system as in (1), a numerical method $\phi_\eta(\cdot)$ is called a symplectic discretization if when applied to a Hamiltonian system, the discrete flow $\phi_\eta : x \to \phi_\eta(x)$ is a symplectic map for sufficient small step sizes. In this paper we focus on the following symplectic discretizations: for $X = (x_1, ..., x_n)^\top, Y = (y_1, ..., y_n)^\top$,

$$Y^{t+1} = Y^t - \eta \nabla_X H(X^t, Y^{t+1}), \quad X^{t+1} = X^t + \eta \nabla_Y H(X^t, Y^{t+1}), \qquad \text{(Type I method)}$$

or

$$X^{t+1} = X^t + \eta \nabla_Y H(X^{t+1}, Y^t), \quad Y^{t+1} = Y^t - \eta \nabla_X H(X^{t+1}, Y^t). \qquad \text{(Type II method)}$$

Both methods are symplectic methods, i.e., they make the map $(X^t, Y^t) \to (X^{t+1}, Y^{t+1})$ to be symplectic. More details of symplectic method can be found in (Haier et al., 2006).

**Cumulative strategies.** In this paper will focus on the dynamics of cumulative strategy and cumulative payoff. The cumulative strategy $X_i(t)$ of agent $i$ is defined as follows :

$$X_i(t) = \int_0^t x_i(s)ds. \tag{3}$$

It is also convenient to write $X_i(t) = X_i(t_0) + \int_{t_0}^t x_i(s)ds$ so the randomness can be introduced into the system at any moment $t_0 > 0$ even $X_i(0) \equiv 0$ in (3).

**Primal-dual correspondence via discretization.** Since we use Euler and Symplectic discretization on the Hamiltonian system, which is not obviously equivalent to the conventionally natural update rules in the strategy spaces $\mathcal{X}_i$, we next establish formally that the Euler or Symplectic discretization of continuous time FTRL with Euclidean norm / negative entropy regularizer implies GDA/MWU or AltGDA/AltMWU respectively. This correspondence can be stated in the following proposition.

**Proposition 3.1.** *For each agent $i \in [2]$, let $\mathcal{X}_i$ denote the strategy spaces, and let*

$$H(X_i, y_i) = h_i^*(y_i(t)) + h_j^*(y_j(0) + A^{(ji)}X_i(t)) \tag{4}$$

*for $j \neq i$ be the Hamiltonian function of (Continuous FTRL), so that the $X_i(t)$ and $y_i(t)$ evolve according to the following Hamiltonian system*

$$\frac{dX_i}{dt} = \frac{\partial H(X_i, y_i)}{\partial y_i}, \quad \frac{dy_i}{dt} = -\frac{\partial H(X_i, y_i)}{\partial X_i}, \tag{5}$$

*Then the following statements holds:*

- *If both players use Euclidean regularizers and $\mathcal{X}_i = \mathbb{R}^{n_i}$, then the Euler discretization of (5) is equivalent to Gradient Descent-Ascent (GDA) on the strategy spaces; and the Symplectic discretization of (5) is equivalent to Alternating Gradient Descent-Ascent (AltGDA) on the strategy spaces.*

- *If both players use entropy regularizers and $\mathcal{X}_i = \Delta_i$ be the simplex, then the Euler discretization of (5) is equivalent to Multiplicative Weights Update (MWU) on the strategy spaces; and the Symplectic discretization of (5) is equivalent to Alternating Multiplicative Weights Update (AltMWU) on the strategy spaces.*

Here equivalent means the variables $(X_i^t, y_i^t)$ getting from the discretizations is the same as the cumulative strategy and payoff of the game dynamics. The Hamiltonian function in the proposition comes from (Bailey & Piliouras, 2019). To prove the correspondence between (AltGDA)/(AltMWU) and Symplectic discretization, we introduce a novel method of discretizing the continuous Hamiltonian system (5) by a combination of two types Symplectic methods, while still keep the symplectic structures on the dynamics of each agents. We believe this method is of independent interests, see Lemma A.2 in Appendix. The detailed proofs of Proposition 3.1 are deferred to Appendix A.

**Random initialization.** We consider the case when noise is introduced to $(X_i(t_0), y_i(t_0))$ at time moment $t_0 > 0$ in continuous FTRL or $(X_i^{t_0}, y_i^{t_0})$ in discrete time settings. The main objective of this paper is to study the evolution of observer uncertainty given the covariance matrix $P_0$ of the initialization $(X_i(t_0), y_i(t_0))$ or $(X_i^{t_0}, y_i^{t_0})$. Take discrete time FTRL for example, the covariance matrix $P_0$ consists of variances $\mathrm{Var}(X_{i,\alpha}^{t_0})$, $\mathrm{Var}(y_{i,\alpha}^{t_0})$, and covariances $\mathrm{Cov}(X_{i,\alpha}^{t_0}, y_{i,\beta}^{t_0})$ for all $\alpha, \beta \in [n_i]$. Tracing the evolution of $\mathrm{Var}(X_{i,\alpha}^t)$ and $\mathrm{Var}(y_{i,\alpha}^t)$ in iterations, we are able to quantify how accurate the prediction will be in FTRL dynamics. Especially we will see in next sections that covariance is a finer measure of observer uncertainty compared to differential entropy.

**Remark 3.1.** *Note that neither the concrete expression nor an approximation of the initial distribution of $(X_i^0, y_i^0)$ is necessary if the historic data is provided (which holds in repeated games), since we can use empirical means and variances as the unbiased estimate of the actual means and variances.*

## 4    DEFICIENCY OF DIFFERENTIAL ENTROPY

Differential entropy, as a measure of observer uncertainty, was used in studying the predictability of MWU in zero-sum games (Cheung et al., 2022). In this section we investigate the evolution of differential entropy in FTRL with differential discretization methods, which correspond to MWU and AltMWU. The main motivation of considering covariance and variance as an alternate measure of observer uncertainty is due to Proposition 4.2, which claims that the differential entropy in alternating update remains **constant**. In other words, differential entropy might be insufficient in capturing the uncertainty evolution for alternating plays.

**Proposition 4.1.** *When two players use (MWU) with step sizes $\eta < \min\{1, 1/\|A\|_2^2\}$, the differential entropy of their cumulative strategy and payoff has linear growth rate, i.e.,*

$$S(X_i^t, y_i^t) \geq S(X_i^{t_0}, y_i^{t_0}) + ct \tag{6}$$

*where $A$ is the payoff matrix of the game, and $c > 0$ is a constant determined by $A$.*

**Proposition 4.2.** *When two players use (AltMWU) with arbitrary step size, the differential entropy of their cumulative strategy and payoff keeps constant, i.e.,*

$$S(X_i^t, y_i^t) = S(X_i^{t_0}, y_i^{t_0}) \tag{7}$$

*for any $t > t_0 > 0$ and $i = 1, 2$.*

The proof of Proposition 4.2 is heavily dependent on the relationship between Symplectic method and (AltMWU), as state in Proposition 3.1. In fact, the evolution of differential entropy is determined by the determinant of the Jacobin matrix of the update rule from $(X_i^t, y_i^t)$ to $(X_i^{t+1}, y_i^{t+1})$, and in each update, the differential entropy is invariant if and only if the absolute value of this determinant equals to 1. As shown in Proposition 3.1, the update rule of $(X_i^t, y_i^t)$ in (AltMWU) constitutes a symplectic map, and the absolute value of the determinant of Jacobin matrix for every symplectic map must equal to 1, therefore we can conclude that the differential entropy in (AltMWU) keeps a constant. The detailed proofs of Propositions 4.1 and 4.2 are deferred to Appendix B, where we also provide numerical examples for these two propositions.

## 5 COVARIANCE IN FTRL

In this section, we are presenting formally the evolution of covariances of cumulative strategies and payoffs. We start with continuous time FTRL with Euclidean regularizers, and proceed in considering Euler and symplectic discretization of continuous time FTRL. In the end, for general FTRL, covariance evolution follows an inequality derived by using techniques of symplectic geometry.

**Covariance evolution in FTRL with Euclidean regularizer.** The evolution of covariance matrix with continuous time FTRL can be deduced from the Hamiltonian formulation of learning dynamics. The Euler discretization of each agent's continuous time FTRL exponentially amplifies the covariance in the learning process. In contrast, symplectic discretization, which has been proven equivalent to alternating update in strategies, amplify covariance of cumulative strategies polynomially and keep that of cumulative payoffs bounded. In this section we will focus on the view point of agent 1, and the same results also hold for agent 2 as they are symmetry.

**Theorem 5.1.** *In two-player zero-sum games, suppose both players use FTRL with Euclidean norm regularizers and unconstrained strategy sets $\mathcal{X}_i = \mathbb{R}^{n_i}$. Suppose at time $t_0 > 0$ the random cumulative strategy and payoff form a random vector $\left(X_i^{t_0}, y_i^{t_0}\right)$ with covariance matrix $P(t_0) \neq 0$. Then for all $t > t_0$ and $\alpha, \beta \in \{1, ..., n_1\}$, the covariance of $(X_i^t, y_i^t)$ evolves in continuous and discrete FTRL according to the following :*

1. *In Euler discretization, for all $\alpha, \beta \in [n_1]$, it holds that $\mathrm{Cov}(X_{1,\alpha}^t, X_{1,\beta}^t)$, $\mathrm{Cov}(y_{1,\alpha}^t, y_{1,\beta}^t)$ and $\mathrm{Cov}(X_{1,\alpha}^t, y_{1,\beta}^t)$ are all of $\Theta(|\mu|^{2t})$ for some number $|\mu| > 1$.*

2. *In continuous time and symplectic discretization, for all $\alpha, \beta \in [n_1]$, it holds that*
   - *if $AA^\top$ is non-singular, then $\mathrm{Cov}(X_{1,\alpha}^t, X_{1,\beta}^t)$, $\mathrm{Cov}(y_{1,\alpha}^t, y_{1,\beta}^t)$ and $\mathrm{Cov}(X_{1,\alpha}^t, y_{1,\beta}^t)$ are of $\mathcal{O}(1)$.*
   - *if $AA^\top$ is singular, $\mathrm{Cov}(X_{1,\alpha}^t, X_{1,\beta}^t)$ is of $\Theta(t^2)$, $\mathrm{Cov}(y_{1,\alpha}^t, y_{1,\beta}^t)$ is of $\mathcal{O}(1)$, and $\mathrm{Cov}(X_{1,\alpha}^t, y_{1,\beta}^t)$ is of $\Theta(t)$.*

Note that the singular conditions on $AA^\top$ can be satisfied by different games in real life, e.g., the payoff matrix of the Rock-Paper-Scissors game makes $AA^\top$ singular, but that of meta-payoff matrix of AlphaStar is non-singular(Czarnecki et al., 2020). Especially, if $A$ is not square, then there must be one player whose $AA^\top$ is singular.

The proof of Theorem 5.1 relies on a detailed analysis on the matrix exponential map associated to solutions of continuous FTRL and matrix powers map associated to solutions of Euler / Symplectic discretize equations, especially in counting the geometric and algebraic multiplicities of 1 as an eigenvalue of the iterative matrices associated to these linear systems. Proofs and additional backgrounds are deferred to Appendix C.

**Covariance Evolution in General FTRL.** Analysis of covariance evolution in FTRL with general regularizer differs from Euclidean regularizer fundamentally. The complexity is mainly caused by the non-linearity of Hamiltonian system induced by continuous time FTRL. Thus, it is inevitable to introduce a localization or linearization scheme of discretization.

So far we have left the evolution of covariance in continuous time FTRL with general regularizers unaddressed. The challenge comes from the non-linearity of Hamiltonian system induced by continuous time FTRL algorithm. Suppose the integral flow of Hamiltonian system is $\phi_t(X_i(t_0), y_i(t_0))$. In the most general setting, we are able to provide a lower bound for the product of standard deviation of $X_{i,\alpha}(t)$ and $y_{i,\alpha}(t)$, i.e., $\Delta X_{i,\alpha}(t)\Delta y_{i,\alpha}(t) \geq$ constant. We state the conditions and results formally in the following proposition.

**Theorem 5.2.** *Let vector $X_i(t)$ and $y_i(t)$ be cumulative strategy and payoff, for $i \in [2]$ and $\alpha \in [n_i]$. Assume that the higher order differentials of $\phi_t(\cdot)$ are bounded by some constant $K$, and the standard deviations $\Delta X_{i,\alpha}(t_0)$ and $\Delta y_{i,\alpha}(t_0)$ at initial time $t_0$ are sufficient small,[2] then for $t > t_0$ it holds that*

$$(\Delta X_{i,\alpha}(t)\Delta y_{i,\alpha}(t))^2 - (\mathrm{Cov}(X_{i,\alpha}(t), y_{i,\alpha}(t)))^2 \geq \frac{1}{2}\frac{w_L^2(P(t_0))}{\pi^2},$$

---

[2]"Sufficietly small" for entries of covariance matrix follows the convention in statistical modeling, (e.g. p166 in (Benaroya et al., 2005).), see Appendix D.3 for more details.

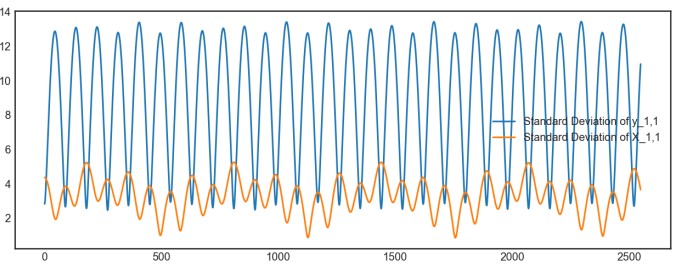

**Figure 1:** The appearance time of the local **minima** of $\Delta(y_{1,1}(t))$ coincide with the local **maxima** of $\Delta(X_{1,1}(t))$, and vice versa. This implies that $\Delta(X_{1,1}(t))$ and $\Delta(y_{1,1}(t))$ can't be small at the same time.

*where $w_L(P(t_0))$ is the linear Gromov width of the ellipsoid defined by the initial covariance matrix $P(t_0)$.*

The definition of Gromov width can be found in Appendix D.2. Note that the inequality holds trivially when there is no uncertainty, i.e., $P(t_0) = 0$, since in this case $w_L^2(P(t_0)) = 0$. The necessary background and details of proof are left in Appendix D. Similar to the *Heisenberg Uncertainty Principle* in quantum mechanics, Theorem 5.2 indicates that $\Delta X_{i,\alpha}(t)$ and $\Delta y_{i,\alpha}(t)$ cannot be small at the same time. An example of FTRL with Euclidean norm regularizers demonstrating this phenomenon is presented in Figure (1).

## 6 EXPERIMENTS

In this section we provide numerical experiments illustrating the covariance evolution results proved for Euclidean norm regularized FTRL in Theorem 5.1. We provide experimental results on the evolution of $\mathrm{Var}(X_{1,1})$, $\mathrm{Var}(y_{1,1})$, the first components of $X_1$ and $y_1$, where $(X_1, y_1)$ evolve as continuous FTRL equation, Symplectic discretization, or Euler discretization. In all experiments, we assume that the payoff matrix is in $\mathbb{R}^{2\times2}$, thus $X_1, y_1 \in \mathbb{R}^2$, and at initial time the covariance matrix $\mathrm{Cov}(y_1, X_1)$ is $[[8,2,1,3],[2,13,7,9],[1,7,9,2],[3,9,2,10]] \in \mathbb{R}^{4\times4}$. More numerical experiments on the non-singular cases are presented in Appendix E. Note that the the y-axis is represented on a logarithmic scale in Figure 2 (a), Figure 3 (a) and Figure 4.

**Continuous time FTRL.** We illustrate how $\mathrm{Var}(X_{1,1}(t))$ and $\mathrm{Var}(y_{1,1}(t))$ evolve with continuous time FTRL with payoff matrices $A_1 = [[1,-1],[-1,1]]$, $A_2 = [[1.2,-1.2],[-1,1]]$, $A_3 = [[1.5,-1.5],[-1,1]]$, see Figure 2. In (a), the $\mathrm{Var}(X_{1,1}(t))$ has a quadratic growth rate, and in (b) $\mathrm{Var}(y_{1,1}(t))$ is bounded, which support results of continuous time part in Theorem 5.1.

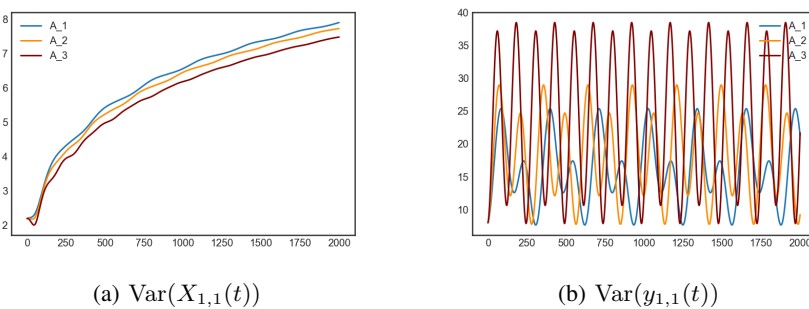

(a) $\mathrm{Var}(X_{1,1}(t))$         (b) $\mathrm{Var}(y_{1,1}(t))$

**Figure 2:** Variance evolution of continuous FTRL, singular cases.

**Symplectic discretization.** We illustrate how $\mathrm{Var}(X_{1,1}^t)$ and $\mathrm{Var}(y_{1,1}^t)$ evolves with symplectic discretization, the payoff matrices are given as follows: $B_1 = [[1,-1],[-1,1]]$, $B_2 = [[1.2,-1.2],[-1,1]]$, $B_3 = [[1,-1.3],[-1,1.3]]$, see Figure 3. From the experimental results, we

can see the variance behavior of symplectic discretization is same as continuous case, which support results of symplectic discretization part of Theorem 5.1.

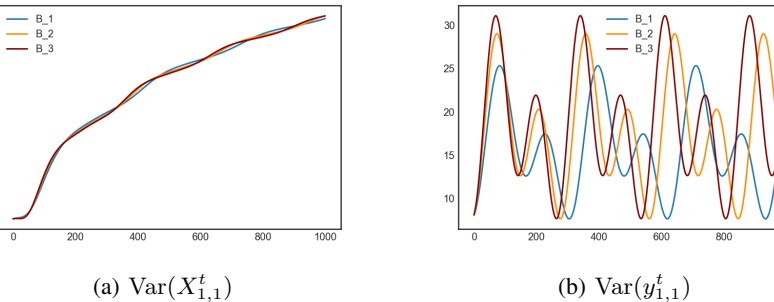

(a) $\mathrm{Var}(X_{1,1}^t)$          (b) $\mathrm{Var}(y_{1,1}^t)$

**Figure 3:** Variance evolution of Symplectic discretization, singular cases.

**Euler discretization.** We show experimental results on $\mathrm{Var}(X_{1,1}^t)$, $\mathrm{Var}(y_{1,1}^t)$ where $(X_1^t, y_1^t)$ evolve as Euler discretization and payoff matrices are given as follows:

- $C_1 = [[1, -1.31], [-1, 1.31]]$ is singular, see (a) of Figure 4.
- $C_2 = [[2, -1.7], [-1.7, 1.5]]$ is non-singular, see (b) of Figure 4.

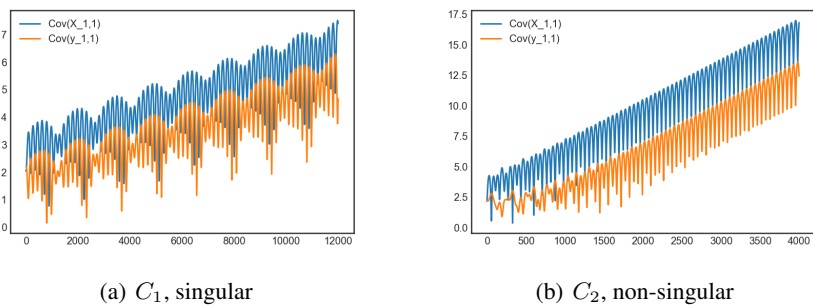

(a) $C_1$, singular          (b) $C_2$, non-singular

**Figure 4:** Variance evolution of Euler discretization.

In Figure 4 we can see that the behavior of $\mathrm{Var}(X_{1,1}^t)$ and $\mathrm{Var}(y_{1,1}^t)$ is independent of whether the payoff matrix is singular or not, and they exhibit an exponential growth rate which support the result of Euler discretization part in Theorem 5.1. Our theoretical results can explain the high frequency oscillations in experimental results. As we shown in Appendix 5.1, the function of the covariance evolution process contains polynomials combinations of trigonometric functions, which cause the oscillations.

## 7 CONCLUSION

In this paper we investigate the evolution of observer uncertainty in learning dynamics from a covariance perspective. We focus on different variants of Follow-the-Regularized-Leader dynamics in zero-sum games and prove concrete rates of covariance evolution for different discretization schemes. Although all such discretization schemes have low regret, alternating discretizations are shown to be far superior from an uncertainty perspective providing a new axis along which to compare learning dynamics in games. In our analysis, we leverage the techniques from symplectic geometry for analyzing the evolution of uncertainty, which to the best of our knowledge is the first of its kind. An interesting direction for future work is to extend this type of analysis for different classes of games (e.g. potential games, mixed motive games, a.o. (Cheung & Tao, 2021; Candogan et al., 2011)) and dynamics.

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
