# OpenReview forum: "Observer Uncertainty of Learning in Games from a Covariance Perspective"
_ICLR.cc/2024/Conference — Submitted to ICLR 2024_

### Official Review · Reviewer_1edF · 2023-10-29

**Soundness:** 3 good
**Presentation:** 3 good
**Contribution:** 2 fair
**Rating:** 6
**Confidence:** 3

**Summary:**

This paper investigates the evolution of observer uncertainty, mainly measured by covariances, in learning dynamics of two-player zero-sum games. The authors focus on the continuous-time FTRL algorithm and its two discretization schemes, Euler and symplectic discretization, which are equivalent to simultaneous and alternating GDA/MWU for certain regularizers. The authors show that for Euclidean regularizer, Euler discretization exponentially amplifies the covariance, while continuous-time FTRL and symplectic discretization amplify the covariance of cumulative strategies polynomially and keep that of cumulative payoffs bounded. As a comparison, the differential entropy of alternating MWU remains constant, which implies that covariance might be a better measurement of observer uncertainty. For general regularizers, the authors establish a Heisenberg-type inequality on variances to demonstrate the tradeoff between strategy spaces and payoff spaces.

**Strengths:**

The strengths of this paper mainly lie in its novelty.
1. The authors propose covariance to measure the observer uncertainty and provide evidence to show that it could be a better measurement than differential entropy.
2. The paper establishes the connection between Euler/symplectic discretization and simultaneous/alternating algorithms through the Hamiltonian system and provides a new perspective to analyze simultaneous and alternating GDA. Moreover, the results demonstrate that the alternating algorithm is more stable than the simultaneous one.
3. Although I do not check the details of the proofs, I believe they are correct.

**Weaknesses:**

I have some doubts about the significance of this paper and there is room for improvement in the presentation.
1. Although the authors claim that the 'input uncertainty' is well-motivated, I am still skeptical about this. For a convergent trajectory, measuring such an uncertainty could tell us (i) how fast we approach the solution; (ii) the initialization would not affect the convergence. However, for a non-convergent trajectory, measuring this uncertainty seems only to tell us whether the trajectory remains bounded or how fast it diverges. Although the results in this paper seem solid, I do not know what are the further implications for these cycling or divergent behaviors.

2. Some results related to cumulative strategies lack interpretation.
The adaptation of cumulative payoffs is easy to understand because we could map it into the strategy space through $\nabla h^*$. However, the adaptation of cumulative strategies seems just to facilitate the definition of the Hamiltonian system. Dividing it by $t$ yields a more intuitive meaning, i.e., the averaged strategies. Consequently, I think that the results in Theorem 5.1 could be transformed into those related to averaged strategies. Then the difference between the covariances of averaged strategies and cumulative payoffs can be better analyzed, because they represent the averaged-iterate and last-iterate behavior, respectively. In particular, I think it is also worth discussing the relationship between the results in Theorem 5.1 and the no-regret property of FTRL as well as the convergence of the averaged strategies. Finally, I do not understand the meaning of the covariance between cumulative strategies and payoffs.

3. The form of symplectic discretization may be not rigorous. If we adopt the symplectic Euler method in [1, Theorem VI.3.3], the update rule of $Y^t$ in symplectic discretization (Type I method) should be $Y^{t+1} = Y^{t} - \eta \nabla_X H( X^t, Y^{t+1} )$, while the authors use $Y^{t+1} = Y^{t} - \eta \nabla_X H( X^t, Y^{t} )$. Although for the Hamiltonian function defined in Proposition 3.1, the two versions are equivalent, I think adopting the latter one may be misleading and less rigorous.

4. The lower bound in Theorem 5.2 involves the covariance, while the amplification rates in Theorem 5.1 only hold for the Euclidean regularizer. For general regularizers, without the amplification rates of the covariance, the result in Theorem 5.2 itself is less meaningful.

Minor concerns
1. The authors could give more intuition on the definition of the Hamiltonian function.
2. The dependence of $\mu$ in Theorem 5.1 on the step size should be clarified.
3. To capture the exponential growth, one could change the y-axis to a log scale. Similarly, to capture quadratic (or more generally, polynomial) growth rate in figures, one could change both axes to log scales.

[1] Ernst Haier, Christian Lubich, and Gerhard Wanner. Geometric Numerical integration: structure preserving algorithms for ordinary differential equations. Springer, 2006.

**Questions:**

1. The results about differential entropy in Section 4 are in terms of MWU and AltMWU, while covariance evolution results in Theorem 5.1 only hold for the Euclidean regularizer, i.e., GDA and AltGDA. They are not directly comparable. However, in the appendix, the authors say that the results in Section 4 hold for more general regularizers. Why not adopt the same regularizer in the main text or state a more general result in Section 4?

2. In Theorem 5.2, whether $AA^\top$ is singular has a significant influence on the rate. Could the authors give a more intuitive explanation of this phenomenon without resorting to the matrix

---

> ### Author Response · Authors · 2023-11-16
> **To reviewer 1edF (1/2)**
>
> We thank you for your careful reading and suggestions in improving the clarity of the paper. We will address your questions and we will be always open for further queries and discussion. Please see our itemized responses below:
>
>  >Motivation of 'input uncertainty'  and further implications for these cycling or divergent behaviors.
>
> Please refer to the second point of the global respond for more discussion on the concept of input uncertainty. As it was shown in previous works, the dynamical behaviors of FTRL is complex, it can exhibit recurrence [1] and chaotic behaviors [2]. However, these properties only provide qualitative descriptions about the hardness of predicting the dynamics of these algorithms. The results in the current paper focus on the quantitative viewpoint, and demonstrates new characteristics of these dynamics, such as the tradeoff between cumulative strategies and payoff. For more discussion on significance, applications if needed see also our related response to Reviewer kXrY.
>
> >The role of cumulative strategies, relationships with convergence of the averaged strategies, and meaning of the covariance between cumulative strategies and payoffs.
>
> Cumulative strategies have a nature game-theoretical explanation: they denote the historical frequencies of strategies used by a player, and they also provide a new coordinate system to describe the dynamics of the two players system. Cumulative strategies and payoffs are related by the identity $y_i(t) = y_i(0) + A^{(ij)} X_j(t)$, for $i,j = 1,2$, thus one player's  cumulative strategies can determine another player's cumulative payoff (although the payoff matrix is unknown), this also explain why we can trace the behaviors of a two players system from the viewpoint of a single player.
>
> We appreciate your suggestion to consider average strategies. Our results on the covariance of cumulative strategies can be used to derive corollaries about the average strategies using the relation $\text{Var}(X(t)/t) = \text{Var}(X(t))/t^2$. For example, in the alternating setting, Theorem 5.1 implies that when $AA^{\top}$ is non-singular, the variance of average strategies tends to $0$ at a rate of $O(1/t^2)$. In case $AA^{\top}$ is singular, the variance of average strategies tends towards a non-zero value, and this number can be determined through computational methods. This suggests that in such cases, the initial distribution will not concentrate as in non-singular cases.
>
> >The form of symplectic discretization.
>
> Thank you for pointing this out, we have fixed this in the update manuscripts. The parts that were modified are  marked by red color.
>
> >The result in Theorem 5.2 itself is less meaningful due to lake of amplification rates of the covariance.
>
> In addition to providing a numerical lower bound, Theorem 5.2 also offers insight into the tradeoff phenomenon between the standard deviations of cumulative payoffs and strategies for more general regularizers. In the case of GDA, where the standard deviations can be calculated precisely, an example of such a tradeoff phenomenon is presented in figure (1) in the main paper. Note that the maximum points of the curve representing standard deviations of cumulative payoffs coincide with  minimum points of the curve representing standard deviations of cumulative strategies, and vice versa. This insight cannot be obtained by analyzing the growth rate of standard deviations of cumulative payoffs and strategies independently since Theorem 5.1 only indicates that both curves are bounded.
>
> >Intuitions on the definition of the Hamiltonian function.
>
> The Hamiltonian function used in the current paper comes from [3], and a similar non-canonical Hamiltonian system formulation (also known as a Possion system) for continuous time  mirror descent algorithms is also presented in [4]. Theorem 5.1 of [3] demonstrates that the Hamiltonian function defined here is inherently connected to the Bregman divergence, which is a commonly used concepts in optimization, plus a an additional term determined by the regularizers and equilibrium of the game.

---

> > ### Author Response · Authors · 2023-11-16
> > **To reviewer 1edF (2/2)**
> >
> > > The dependence of $\mu$ in Theorem 5.1 on the step size.
> >
> > As shown in the proof (Appendix C.2.2 ), the constant  $\mu$ in Theorem 5.1 has norm $\lvert \mu \lvert = 1+\gamma \eta^2$, where $\eta$ is the step size and $\gamma$ is the maximal eigenvalue of $AA^{\top}$.
> >
> > > Problems with figures.
> >
> > Thank you for the suggestion, we have changed the y-axis to a log scale in the update manuscripts. Please refer to the new manuscripts in the system.
> >
> > > Why not adopt the same regularizer in the main text or state a more general result in Section 4?
> >
> > We present the results in Section 4 using MWU and AltMWU to enable a direct comparison with the results in [2], where differential entropy is used as a measure of uncertainty and the differential entropy of MWU is calculated. However, the proof of our results in Section 4 doesn't rely on any special choice of regularizer.
> >
> > > In Theorem 5.2, whether $AA^{\top}$ is singular has a significant influence on the rate. Could the authors give a more intuitive explanation of this phenomenon without resorting to the matrix.
> >
> >
> > Thank you for asking this important question. We think the phenomenon can be intuitively explained in terms of the covariance evolution of the average strategies. As shown in [5], in an unconstained min-max problem, strating from a fixed initial point, the average strategies of AltGDA will converge to an equilibrium. When $AA^{\top}$ is non-singular, for example, when $A$ is an invertible square matrix, the only equilibrium is the all zero vector, and the average strategy converge to this point for all initial point, thus in this case the covariance of average strategy is small, this implies cumulative strategy growth with a slow rate. However, when $AA^{\top}$ is singular, for example, when $A$ is an non-invertible square matrix, there are several different equilibria, and the average strategy will converge to these different equilibria depending on the initial point, thus during the process the covariance of average strategy is still large, this leads to a large covariance of the cumulative strategy.
> >
> > We hope these can address your concerns, and we are always open to future discussions.
> >
> > References:
> >
> > [1] Mertikopoulos et al., Cycles in adversarial regularized learning
> > , SODA 2018
> >
> > [2] Cheung et al., Chaos, Extremism and Optimism: Volume Analysis of Learning in Games, NeurIPS 2020
> >
> >
> > [3] Bailey et al., Multi-agent Learning in Network Zero-Sum Games is a Hamiltonian System, AAMAS 2019.
> >
> > [4] Wibisono et al., Alternating Mirror Descent for Constrained Min-Max Games, NeurIPS 2022.
> >
> > [5] Gidel et al, A variational inequality perspective on generative adversarial networks, ICLR 2019.

---

> > > ### Comment · Reviewer_1edF · 2023-11-21
> > > **Thanks for the response**
> > >
> > > Thank the authors for their detailed response. I have no further questions now and I keep my score.

---

> > > > ### Author Response · Authors · 2023-11-21
> > > >
> > > > Thank you for your support!

---

### Official Review · Reviewer_kXrY · 2023-10-29

**Soundness:** 3 good
**Presentation:** 3 good
**Contribution:** 2 fair
**Rating:** 5
**Confidence:** 3

**Summary:**

This paper studies the evolution of uncertainty or observation accuracy in game dynamics by characterizing the growth rate of a certain covariance information. In particular, they focus on two-player zero-sum games and the continuous-time FTRL with different regularizers, as well as two discretization methods, namely Euler and sympletic. The theoretical results reveal that the symplectic discretization improves the accuracy of prediction in game dynamics, which is also confirmed experimentally.

**Strengths:**

The paper provides a novel characterization of continuous-time FTRL and discretizations thereof in two-player zero-sum games. The new results are also connected to earlier works (such as Cheung and Piliouras (2020) and Cheung (2022)). The separation in the behavior of  Euler versus sympletic discretization is also conceptually interesting. Furthermore, the results are non-trivial from a technical standpoint, combing tools from different areas. It is possible that such techniques could be of independent interest for future work in this area. The results appear to be sound; I did not find any notable issue.

In terms of the presentation, the writing overall is clear, and the key ideas are carefully explained. It was generally easy to follow the paper.

**Weaknesses:**

The main issue I have is with regards to the motivation and the significance of the results. Although the authors already attempt to discuss about the motivation in quite length in the introduction, I still cannot see any concrete motivation or applications for those results. Overall, the paper provides throughout several facts about the game dynamics but without explaining the significance of the characterization. Can the authors provide some actual applications where the new results can be relevant? Otherwise the results appear to some extent artificial.

**Questions:**

A couple of minor stylistic issues:

1. Footnotes should come after punctuation marks
2. The references are not used appropriately; for example Go Silver et al. (2016) should instead be Go (Silver et al.)

---

> ### Author Response · Authors · 2023-11-16
> **To reviewer kXrY**
>
> We thank you for your careful reading and suggestions in improving the clarity of the paper. We will address your questions and we will be always open for further queries and discussion. Please see our itemized responses below:
>
> >Motivation, significance of the results, and actual applications.
>
> Please refer to the global respond for more explanation of the motivation.
>
> For the significance : Game dynamics particularly based on zero-sum like game are seminal components in standard AI architectures such as GANS [1] (7000+ refs), PSRO [2] (600+ refs) and are actually useful even in n-player games (Diplomacy [3]) The typical way to control such dynamics so far has been to aim/hope for global convergence to equilibrium, however, there are many hurdles to this tasks both from a computational perspective [4,5] as well as from a dynamical systems perspective [6]. Here we aim at a different type of "stability" altogether. Stability/predictability of orbits *without* needing to converge to Nash as a necessary intermediary step from proving stability. By showing that this is possible we open the door for new multi-agent AI architectures that although maybe design around the pipe-dream of global  convergence they can still be practically useful.
>
> Actual applications: (Computer Poker). Poker is of course a standard benchmark for AI and a real world application.  This direction of work could provide some insights behind the empirical success of CFR+ [7,8] in poker. CFR+ is a variant of Counter Factual Regret (CFR) minimization [9] with improved empirical performance. A key difference between CFR+ and vanilla CFR is switching from simultaneous updates to alternating updates. Our results provide some hints on why this could lead to increased performance. Increased orbit stability, reduced payoff variances due to symplectic-like discretization would lead to faster time average convergence to equilibrium and thus improved performance. Of course our current work is not on CFR and extensive form games but normal form games, nevertheless, we believe these are promising steps that provide a new vocabulary to study such real-world systems.
>
> >Minor stylistic issues :
>
> Thank you for pointing out these issues. We have fixed these issues and updated the manuscript in the system. Footnotes are moved after punctuation marks, and the references have been unified in the (authors,year) format. Thank you again!
>
>  We hope these can address your concerns, and we are always open to future discussions.
>
> References:
>
> [1] Goodfellow, Ian, et al. "Generative adversarial networks." Communications of the ACM 63.11 (2020): 139-144.
>
> [2] Lanctot, Marc, et al. "A unified game-theoretic approach to multiagent reinforcement learning." Advances in neural information processing systems 30 (2017).
>
> [3] Meta Fundamental AI Research Diplomacy Team (FAIR)†, et al. "Human-level play in the game of Diplomacy by combining language models with strategic reasoning." Science 378.6624 (2022): 1067-1074.
>
> [4] Daskalakis, Constantinos, et al. "The complexity of computing a Nash equilibrium." Communications of the ACM 52.2 (2009): 89-97.
>
> [5] Daskalakis, Constantinos, et al. "The complexity of constrained min-max optimization." Proceedings of the 53rd Annual ACM SIGACT Symposium on Theory of Computing. 2021.
>
> [6] Milionis, Jason, et al. "An impossibility theorem in game dynamics." Proceedings of the National Academy of Sciences 120.41 (2023): e2305349120.
>
> [7] Oskari Tammelin. Solving large imperfect information games using cfr+. arXiv preprint
> arXiv:1407.5042, 2014
>
> [8]    Michael Bowling et al. Heads-up limit holdem
> poker is solved. Science, 347(6218):145–149, 2015.
>
> [9]  Martin Zinkevich et al. Regret minimization in games with incomplete information. In Advances in neural information processing systems, pages 1729–1736, 2008.

---

> > ### Comment · Reviewer_kXrY · 2023-11-19
> > **Thank you for the Response**
> >
> > I thank the authors for their response. I agree with the authors that understanding the effectiveness of alternating dynamics is an important question, which is currently poorly understood. My main concern at the moment is that there is a considerable gap between the results and the applications mentioned above, for example understanding CFR+ in poker. But the paper still makes an interesting step towards a better understanding of alternating dynamics. I have no further questions at the moment.

---

> ### Author Response · Authors · 2023-11-21
> **About actual applications**
>
> Thank you for your replying and interests to the application of our results. We have realized that the concern is about the specific application scenarios of our results. We guess the "gap" might refer to two possibilities, and we explain on both of them. If we have missed the point, please leave comments and we are happy to provide futher information.
> 1. If the gap means the model we consider and the applications mentioned, then the actual applications follows:
>
>     First of all, we consider zero-sum games which is a model of multi-agent systems in AI, our results serves directly in any scenarios where zero-sum matrix games are involved.
>
>     Secondly, in areas like GANs, theory of zero-sum games are studied as a simlified case to provide theoretical explanation to phenomenon from GANs (and other computer games), or provide theoretical guidence on algorithm designs.
>
>     Thirdly, understanding algorithms in bilinear matrix games is a first and fundamental step towards general setting that actual captures models in real world. Recall that the standard approach on learning algorithms in min-max optimization is the following:
>     $$\text{zero-sum matrix game=>convex-concave game=>non-convex non-concave game},$$
>    This is why we mentioned that current results have given promising steps on future works.
>
> 2. If the gap refers to the application of variance/covariance, we argue that
>     The real-life application of variance/covariance is often tied with concentration inequalities. Let's take Chebyshev inequalities for example $$Pr(|X>\mu|>k\sigma)<\frac{1}{k^2}.$$
> By results of Theorem 5.1, we know that $\text{Var}X^t_i$ is of $\Theta(t^2)$. Say $\text{Var}X^t_i=ct^2$ (the formulation of $\text{Var}X^t_i$ can be explicitly determined according to our analysis process in Appendix C.2.3, using the information provided by the payoff matrix and initial conditions), and then we have
> $$\text{Pr}(|X^t_i-\mu|>k\sqrt{c}t)<\frac{1}{k^2}$$
> for any $k>1$. Therefore, we have concrete estimate of the probability for $X^t_i$ to deviate from the mean at **Each Iterate**, which captures the accuracy of prediction in each iterate. For the cumulative payoff $y^t_i$, the alternating play keeps variance bounded, i.e., $\text{Var}y^t_i=\mathbb{E}|y_i^t-\mu|^2\le c$ and this implies (using Markov inequality)
> $$\text{Pr}(|y^t_i-\mu|>k)<\frac{c}{k^2},$$
> which also captures the accuracy of prediction of payoff, e.g., the probability of $y^t_i$ to be greater than $\mu+10$ or less than $\mu-10$ is less than $c/100$.
>
> Thank you again for your feedback. Please let us know whether we have addressed the "actual application" concern or further clarifying is needed.

---

> ### Author Response · Authors · 2023-11-22
> **Have we addressed your concerns?**
>
> Dear reviewer kXrY,
>
> Since the discussion time is coming to an end, may I ask if our response have solved your concerns? Please let us know if further clarifying is needed, and we are always open to provide additional discussion. If our answers have resolved your concerns, we would greatly appreciate it if you could consider updating your recommendation accordingly. Thank you again for your feedback !

---

### Official Review · Reviewer_3XfB · 2023-11-04

**Soundness:** 3 good
**Presentation:** 3 good
**Contribution:** 3 good
**Rating:** 6
**Confidence:** 3

**Summary:**

The authors study the observer uncertainty in deterministic learning dynamics of zero-sum games with random initializations. They explore the follow-the-regularized-leader (FTRL) algorithm in two-player zero-sum games and analyze its continuous-time as well as Euler and symplectic discretization dynamics. They give bounds on the growth rate of the covariance variables (cumulative payoff and cumulative strategy) during evolution for CT, Euler and symplectic cases for L2 and negentropy regularizations. They also establish a Heisenberg-type uncertainty inequality for variances of variables under CT, Euler and symplectic dynamics under general regularizers. Furthermore, they demonstrate by analysis and numerical experiments that symplectic discretization improves the accuracy of prediction in learning dynamics.

**Strengths:**

The uncertainty inequality for general regularizers and its connection to Hamiltonian systems seem interesting and original. Motivation, objevtives and results are stated clearly.. Theoretical results are also easy to follow.

**Weaknesses:**

related works can be discussed more comprehensively.  the limitations of the results and assumptions can be eloborated more to make it easier to follow.

**Questions:**

I don't have any questions to authors.

---

> ### Author Response · Authors · 2023-11-16
> **To reviewer 3XfB**
>
> Thank you for your supportive comments and interest to our work! We have added more discussions on related works, please see our global respond. We will be always open for further queries and discussion.

---

### Official Review · Reviewer_snif · 2023-11-05

**Soundness:** 3 good
**Presentation:** 3 good
**Contribution:** 2 fair
**Rating:** 6
**Confidence:** 3

**Summary:**

This paper investigate the evolution of observer uncertainty in the Follow-the-Regularized-Leader dynamics for learning to solve zero-sum games. The authors prove concrete rates of covariance evolution for different discretization schemes. The proofs rely on the techniques from symplectic geometry for analysing the evolution of uncertainty.

**Strengths:**

The authors provide rigorous, theoretical analysis for the uncertainty in the Follow-the-Regularized-Leader dynamics for learning to solve zero-sum games. The proofs are given in detail. The paper is well written.

**Weaknesses:**

I would say this paper is not well-motivated. I appreciate the mathematical rigour of the theory, but can hardly see how it is useful for machine learning. The authors are needed to clarify why we need to study this problem, and how the results would contribute to the community in the context of machine learning (such as understanding algorithms or designing new algorithms?)

I am also concerned about the novelty. The first 5 pages do not include new theory. Section 4 seems directly from Cheung et al. (2022). The proofs for Section 5 heavily rely on existing results - a very large proportion of the "proofs" are actually quoting existing papers, and the new part seems actually combining existing lemmas/propositions, or simple matrix calculations. Please clarify.

I was excited to see the abstract talks about Heisenberg Uncertainty Principle, but later found little theory is really relevant - please consider removing "Heisenberg Uncertainty Principle" or giving more discussion.

**Questions:**

Please address the above.

---

> ### Author Response · Authors · 2023-11-16
> **To reviewer snif (1/2)**
>
> Thank you for your careful reading and suggestions in improving the clarity of the paper.  We will clarify the issues in revision. Please see our itemized responses below:
>
> >This paper is not well-motivated.
>
> Please refer to our global respond.
>
> >“I am also concerned…simple matrix calculations.”
>
> We are trying to address this comment but we found the descriptions are too vague. It would be great if the reviewer can make the points specific so that we can address the concerns. To elaborate:
>
> >"The first 5 pages do not include new theory."
>
> Yes, the first 5 pages are devoted to explaining the setting, motivation, and background material related to our work. Since we are combining ideas, tools and techniques from different fields such as online learning (e.g. GDA/MWU/AltMWU), game theory (strategies/zero-sum games), dynamical systems (Hamiltonians), information theory (differential entropy) and statistics (covariance), we clearly need some space to establish the necessary vocabulary for the reader.  We believe that given the breadth of the ideas explored, we are actually using space rather efficiently. If the reviewer believes otherwise, we will be happy to hear some concrete suggestions about which parts are superfluous and should be cut out.
>
> >"Section 4 seems directly from Cheung et al. (2022)."
>
> This is not true. Both the theoretical results and proof strategies of section 4 in the current paper are different from Cheung et al. (2022).
>
> Firstly,  Cheung et al. (2022) didn't consider how differential entropy evolve in the alternating play setting. In Proposition 4.2, we prove
> the differential entropy keeps constant in the alternating play setting, thus the concept of differential entropy can not capture the uncertainty evolution in the alternating play setting. This also motivate us to seek other measure of uncertainty evolution.
>
> Secondly, the proof by Cheung et al. (2022) involves a complex calculation on the Jacobian of the multiplicative weights update algorithm, which is difficult to generalize to the alternating play setting due to its requirement for two strategies updating. Our proof of Proposition 4.2 simplifies this kind of calculation by establishing a connection between symplectic discretization and the alternating play setting (Proposition 3.1). Please refer to the discussion below Proposition 4.2 in the paper for more detailed explanations on the differences of the proof strategies.

---

> > ### Author Response · Authors · 2023-11-16
> > **To reviewer snif (2/2)**
> >
> > >"The proofs for Section 5 heavily rely on existing results - a very large proportion of the "proofs" are actually quoting existing papers, and the new part seems actually combining existing lemmas/propositions, or simple matrix calculations."
> >
> > This is not true either. There are important new technical ideas with no analogues in prior work in learning in games. For example, a totally novel element is the use of the deep and celebrated insights related to Gromov's symplectic rigidity, see (https://mathoverflow.net/questions/355928/what-is-symplectic-rigidity).  Can you please provide a reference to any work on learning in games or even ML more generally utilizing this idea?
> >
> > Going into more details in regards to our technical contributions, there are two theorems in Section 5. We will summarize our proof strategies and highlight several novel aspects comparing to other papers.
> >
> > Firstly, in Theorem 5.1, we state results on the evolution of covariance matrix in  Euclidean norm regularizers. The proof (Appendix C.2.) relies on a calculation of the growth rate of elements in the relevant exponential or power maps that appear in the algorithm. We achieve this by using the tool of Jordan normal forms. To make the proof self-contained, we provide basic backgrounds in Appendix C.1. To calculate the Jordan normal form, we need to know
> >
> > **(1) eigenvalues, (2) the number of Jordan blocks of each eigenvalues and (3) the size of each Jordan blocks**.
> >
> > Matrix analysis naturally also appear in several other papers that study the behaviors of game dynamics in terms of their last iterations, such as [1], [2], and [3]. However, in contrast to our work, for the purpose of studying last-iterate behaviors, it is enough to know the spectral radius of the involved matrices, thus only the aspect of **(1) eigenvalues** is involved in these papers. However, to achieve our goal, we also need to analyze the aspects of (2) and (3), a large part of Appendix C.2 is devoted to this task, and we believe that these parts are far from trivial.
> >
> > Secondly, the technical novelty of Theorem 5.2, is the application  of Gromov's symplectic rigidity (Theorem D.1) in online learning and game theory. To our best knowledge, this is the first attempt to use morden symplectic geometry to analyze the behavior of FTRL. Gromov's theory has been used by Maurie A. de Gosson to derive Heisenberg's inequalities with covariance terms. (The Symplectic Camel and the Uncertainty Principle:The Tip of an Iceberg? Found Phys, 2009.) Different from the techniques of de Gosson, we use methods in H. Benaroya, S. Han, and M. Nagurka to obtain the inequality of Theorem 5.2. Actually the method of de Gosson is not valid in our setting because we don't consider "quantum" and there is no Planck constant (which plays a essential role in de Gosson's original proof). Of course understanding this bibliography, identifying the connections to our setting and adapting it to our work is far from a trivial task.
> >
> > >More discussions on the Heisenberg Uncertainty Principle.
> >
> > Thank you for your interests on this topic and we are happy to add more discussion on the relation between Heisenberg uncertainty and our results.
> >
> > Heisenberg uncertainty is in its essence an inequality of the form $\Delta p\Delta q\ge h$, where $q$ and position and $p$ is momentum. In FTRL, the cumulative strategy $X$ corresponds to position $q$ and cumulative payoff $y$ corresponds to momentum $p$. Our paper is not about Heisenberg uncertainty or using it in our results, instead our results have precisely the same flavor as Heisenberg's inequality, e.g., Theorem 5.2. says the standard deviation of payoff and strategy cannot be simultaneously small, which corresponds to Heisenberg’s inequality implying standard deviations of momentum and position cannot be small simultaneously. We also note that other reviewers do not mention that it is inappropriate to informally point out this kind of relationship.
> >
> > References :
> >
> > [1] Daskalakis et al., Training GANS with  optimism, ICLR 2018.
> >
> > [2] Zhang et al., Convergence of gradient methods on bilinear zero-sum games, ICLR 2020.
> >
> > [3] Gidel et al, A variational inequality perspective on generative adversarial networks, ICLR 2019.

---

> > > ### Comment · Reviewer_snif · 2023-11-22
> > >
> > > Thanks for your response. My previous concerns are largely clear. The "global" response needs to be clearly included in your paper. A discussion on the novelty and significance of your proofs is also needed. I'm willing to raise my score.

---

> > > > ### Author Response · Authors · 2023-11-22
> > > >
> > > > Thanks a lot for your suggestions and support! It is great that we have addressed your concerns. We are adding the global response and other remarks based on the comments to the revision.

---

### Author Response · Authors · 2023-11-16
**For all reviewers**

We thank reviewers for all the helpful comments. We would like to explain the motivation of the current paper regarding the common concern of several reviewers.


>(1). The place of this paper in the literature and the motivation.

The current paper is situated within the research on the dynamics of no-regret online learning algorithms and represents a natural extension of previous works that address this problem from the perspectives of volume or uncertainty evolution, such as [1], [2], and [3]. A main motivation for this work  is to address certain drawbacks in these works and further expand upon their results. The following will provide a more detailed explanation.

In [1] and [2], volume analysis was introduced as a tool to study the evolution of a region of initial conditions under the simultaneous MWU algorithm. From this perspective, it was discovered that the simultaneous MWU algorithm exhibits what is known as Lyapunov chaos behaviors. Similar tools are also used to study day-to-day behaviors of no-regret learning algorithms in [4]. Driven by the need of considering the initial uncertainty of learning in game (see the second point in this global respond), [3] purposed to use differential entropy  as a measurement of uncertainty, and extended volume analysis to calculate how differential entropy evolve under simultaneous MWU algorithm.

On this stage, it is nature to ask :

**Is differential entropy a good measure of uncertainty in games dynamics? More especially, as alternating playing is more common in real word games such as chess, card game and training of GANs, can the results in [3] be extended from simultaneous playing to alternating playing ?**

Motivated by the above questions, we have found that differential entropy is not a suitable measure for capturing changes in uncertainty during alternating play. Therefore, the next question is:

**Which concept is more appropriate for measuring uncertainty evolution in game dynamics?**

The idea of using variance as a measure of uncertainty originates from the Heisenberg uncertainty principle, where covariance and standard deviation are employed to quantify the uncertainty in the quantum dynamics. Moreover, since we care about day-to-day behavior and the risk in decision making, variance and covariance has prominent advantage in quantifying risk (it has been well studied in statistics and finance, data science), this is why we study uncertainty with variance and covariance.

>(2). Where is the uncertainty coming from in learning in games?

“Input uncertainty” is inevitably introduced in decision making, agents’ initial beliefs can be affected by irrationality of players, incomplete information, mechanism of game design (e.g., bandit feedback, stochastic oracle feedback). It is also inescapable in all computer simulations due to finite accuracy of the representation and rounding errors. Though such input errors may be small if their effects blow up quickly, they cannot be safely ignored. On the other hand, training generative adversarial networks is usually understood as a learning problem in zero-sum games, and the input uncertainty refers to the initialization of neural network based on some predetermined distribution.

References :

[1] Cheung et al., Vortices Instead of Equilibria in MinMax Optimization: Chaos and Butterfly Effects of Online Learning in Zero-Sum Games, COLT 2019

[2] Cheung et al., Chaos, Extremism and Optimism: Volume Analysis of Learning in Games, NeurIPS 2020

[3] Cheung et al., The Evolution of Uncertainty of Learning in Games, ICLR 2022

[4] Flokas et al., No-regret learning and mixed Nash equilibria: They do not mix
, NeurIPS 2020

---

### Author Response · Authors · 2023-11-23
**New version of manuscript has been updated**

Dear reviewers,

Thank you for your comments on increasing the clarity of the current paper. We have updated a new version of the manuscript according to your comments. The modified parts have been marked in red color. To summarize: we have added more detailed explanations on the motivation and novelties of technologies, provided more discussion on the related work, and fixed several minor errors pointed out by reviewers. We are happy to provide further discussion if you have any other concerns. Thank you again for all your hard work!

---

### Meta-Review · Area_Chair_uaak · 2023-12-05

**Metareview:**

This paper addresses measuring uncertainty in game dynamics. The authors show that differential entropy is not, but variance is. The reviewers highlighted opportunities to clarify the motivation and novelty of the results. The authors wrote extensive replies to those concerns and included additional text to address the issues. That additional text harbors some grammatical issues, but it is expected that they will be fixed prior to publication. The motivation and utility of the genesis of the variance from quantum physics may need some support - it seems one could propose variance as a measure of uncertainty without appealing to quantum physics - but the end result seems to be of primary focus in this work, not the path that led to it. The authors show the performance of their proposed methodology in experiment, but do not seem to compare to any other methods or a baseline and do not seem to analyze the computational aspects of their method.

**Justification For Why Not Higher Score:**

The reviewers were concerned about the applicability of the paper to machine learning problems and the novelty of the method compared to the main preceding work by Cheung. There were no comparisons to other methods or computational aspects of the work.

**Justification For Why Not Lower Score:**

N/A

---

### Decision · Program_Chairs · 2024-01-16

Reject